# Psoas Abscess Due to *Mycobacterium avium* in a Patient with Chronic Lymphocytic Leukemia—Case Report and Review

**DOI:** 10.3390/jcm8020216

**Published:** 2019-02-07

**Authors:** Natascha D. Diaco, Bettina Strohdach, Anna L. Falkowski, Nicolin Hainc, Philippe Brunner, Jonas Rutishauser, Lorenz Jost, Philip E. Tarr

**Affiliations:** 1University Department of Medicine, Kantonsspital Baselland, University of Basel, 4101 Bruderholz, Switzerland; natascha.diaco@stud.unibas.ch (N.D.D.); Bettina.strohdach@usb.ch (B.S.); j.rutishauser@unibas.ch (J.R.); lorenz.jost@ksbl.ch (L.J.); 2Infectious Diseases Service, University Department of Medicine, Kantonsspital Baselland, University of Basel, 4101 Bruderholz, Switzerland; 3Department of Radiology and Nuclear Medicine, Kantonsspital Baselland, University of Basel, 4101 Bruderholz, Switzerland; anna.falkowski@usb.ch (A.L.F.); nicolin.hainc@usz.ch (N.H.); 4Neuroradiology Service, University Hospital Zurich, University of Zurich, 8091 Zurich, Switzerland; 5Cantonal Institute of Pathology, 4410 Liestal, Switzerland; philippe.brunner@viollier.ch; 6Pathology Service, Viollier AG, 4055 Basel, Switzerland; 7Department of Medicine, Kantonsspital Baden, 5404 Baden, Switzerland; 8Oncology Service, Kantonsspital Baselland, University of Basel, 4101 Bruderholz, Switzerland

**Keywords:** *Mycobacterium avium*, abscess, infection, chronic lymphocytic leukemia, non-tuberculous mycobacteria, obinutuzumab, ibrutinib, immunosuppression

## Abstract

Infections may constitute a serious complication in patients with chronic lymphocytic leukemia (CLL). New treatment agents including obinutuzumab and ibrutinib have improved the progression-free survival in CLL, and data suggest a similar overall infection risk and a limited risk of opportunistic infections when compared to standard chemo-immunotherapy. Nevertheless, cases of opportunistic infections including non-tuberculous mycobacterial (NTM) in CLL patients have recently been published. We present a case of a 74-year old man with extensive prior CLL treatment history, including most recently obinutuzumab. He developed an abscess of the psoas muscle and inguinal lymphadenopathy. An inguinal node biopsy specimen showed infection with *Mycobacterium avium*, confirmed by broad-spectrum mycobacterial PCR, *M. avium*-specific PCR, and mycobacterial culture. This case and our literature review suggest that physicians should be aware of opportunistic infections in patients with CLL. Diagnostic differentiation from CLL disease progression, Richter’s transformation to aggressive lymphoma, and secondary malignancy relies on histological and appropriate microbiological studies from biopsy material of affected organs. Infection prophylaxis in CLL should be considered, including vaccinations and intravenous immune globulin replacement.

## 1. Introduction

Infection with *Mycobacterium avium* complex (MAC) is a rare complication in patients with chronic lymphocytic leukemia (CLL), with four previous cases reported in the literature [1,2,3,4]. Here we present a patient with CLL who developed a psoas abscess due to *M. avium*, after having received extensive previous CLL therapies and five months after first receiving treatment with obinutuzumab. This case and our literature review suggest that physicians should be aware of opportunistic infections in patients with CLL. Accurate diagnosis of progressive lymphadenopathy in CLL requires appropriate biopsy material to be analysed microbiologically and histologically.

## 2. Case Report

A 60-year old man was diagnosed with CLL. He underwent extensive courses of CLL treatment, which included chlorambucil, prednisone, fludarabine, rituximab, cyclophosphamide, bendamustin, ofatumumab, and lenalidomide.

Nine years later, hypogammaglobulinemia and recurrent respiratory infections were noted, and monthly intravenous immune globulin (IVIG) infusions were started. Four years later, progressive generalized lymphadenopathy with bulky retroperitoneal masses, weight loss and malaise were attributed to progressive CLL, and monthly infusions of obinutuzumab were given. Five months later, the patient presented with edema and pain of his right leg, along with painful swelling in the right groin despite regression of the generalized lymphadenopathy. Computed tomography (CT) showed a significant increase of the retroperitoneal lymph node masses and a new, contrast-enhancing fluid collection in the right psoas muscle extending to the groin (Figure 1). Bacterial culture of a CT-guided inguinal node biopsy specimen remained sterile, while broad spectrum mycobacterial PCR, *M. avium*-specific PCR, and mycobacterial culture were all positive for *M. avium*. Histological examination showed necrotic, histiocyte-predominant inflammation with numerous acid-fast bacilli but no evidence of Richter’s transformation to an aggressive lymphoma. External catheter drainage of the abscess was done for 8 weeks. CLL treatment was interrupted.

The isolate was susceptible in vitro to clarithromycin and rifampicin (minimal inhibitory concentrations, 4 mg/L and 20 mg/L, respectively). Therapy with rifampicin 600 mg/day, ethambutol 1000 mg/day and clarithromycin 500 mg/day was given. The second cycle of obinutuzumab was delayed by three months due to the infectious complication. The third cycle was started on time but was terminated early due to progression of the CLL. Eight months into *M. avium* therapy, complete remission of the abscess without any relapse were noted on CT. However, generalized lymphadenopathy re-appeared. *M. avium* treatment with rifampicin, ethambutol, and clarithromycin was continued and ibrutinib was started, which again led to a major response of the lymphadenopathy. The initial dose of ibrutinib was reduced to 280 mg due to low blood counts and the risk of potential drug interactions with increased blood levels of ibrutinib. The second and third cycles of ibrutinib were started with 420 mg for 2 weeks and reduced to 280 mg for the rest of the cycle due to low platelet counts. Later cycles were given at the standard dose for CLL, i.e., 420 mg without evidence of enhanced hematotoxicity despite the combination with the antimycobacterial agents. Blood level measurements for ibrutinib were not done.

Unfortunately, the patient died 10 months after initiating antimycobacterial treatment, in the setting of massive pleural hemorrhage and bleeding into the mediastinal lymph nodes. At autopsy, no macroscopic or histological evidence of the infection with *M. avium* was found in the area of the original abscess or elsewhere.

## 3. Literature Review and Discussion

Lymphadenopathy in patients with CLL has a broad differential diagnosis, which most commonly includes progressive or treatment-unresponsive CLL, but also Richter’s transformation to aggressive lymphoma, second malignancies, and infectious lymphadenitis [4]. When adenopathy is not pronounced and the patient is well, without systemic symptoms or cytopenias, successful treatment with a course of antibiotics (without making a specific diagnosis) has been recorded [4]. However, when lymphadenopathy is progressive or generalized, or when the patient is systemically ill, accurate diagnosis becomes important, and typically requires lymph node biopsy material to be sent for histological and microbiological (including mycobacterial) studies. 

### 3.1. Infections with Non-Tuberculous Mycobacteria (NTM)

NTM are ubiquitous in the environment, are not considered to be transmitted from person to person, and their incidence seems to be increasing in several geographic locations [5,6,7]. The lungs are the most common site of NTM infections, with 90% of positive NTM cultures derived from pulmonary secretions. Approximately half of these cases are considered true infection and half are attributed to respiratory tract colonization [5,6,8]. M. avium complex (MAC) is by far the most common NTM species recovered from the lungs. Immunosuppressive treatment is recorded in approximately 25% of patients with NTM infections [8], and in immunocompromised individuals, extrapulmonary and disseminated NTM infections may commonly occur [8,9]. In NTM patients who are not receiving immunosuppressive treatment, common predisposing conditions may include chronic obstructive pulmonary disease and bronchiectasis [5,8].

There are four previously reported cases of MAC infection in patients with CLL in the literature (Table 1). These reports have included patients with various prior CLL therapies who developed MAC lymphadenitis [1], small bowel infection [2], or disseminated cutaneous disease [3] (Table 1). One patient developed recurrent MAC lymphadenitis that was first noted before any CLL therapy was given [4].

Clinical recognition of MAC infection can be difficult. In our patient, progressive CLL or Richter’s transformation to an aggressive lymphoma was initially suspected as the cause of increasing retroperitoneal lymphadenopathy. MAC infection was only diagnosed when the newly appearing groin mass was biopsied; mycobacterial cultures were only requested after infectious diseases consultation was obtained, when standard cultures of the biopsy material remained without bacterial growth.

In addition to MAC, other NTM infections have been recorded in CLL patients, including infections with M. chelonae [10,11,12], M. marinum [13,14], M. szulgai [15], M. fortuitum [16] and M. genavense [17] (Table 2). These patients were affected mostly by disseminated cutaneous lesions (*n* = 6, [10,11,12,13,14,15]) or nodular pulmonary infections (*n* = 2, [13,16]). One patient presented with multifocal osteomyelitis and skin lesions [15].

### 3.2. Infectious Complications in CLL

Infectious complications are well-recorded in patients with CLL and may be responsible for more than half of all deaths in these patients [18,19,20,21]. CLL patients also have an increased incidence of second malignancies, suggesting impaired immune surveillance [18]. Infections in CLL have been attributed to several mechanisms, which may coexist in an individual patient, including patient factors (increasing age, comorbidities), and inherent immune dysfunction related to CLL itself (hypogammaglobulinemia, dysfunction of T-cells, NK-cells, and dendritic cells, complement defect, phagocyte dysfunction) [18,22]. The relevance of such inherent immunological dysfunction in CLL is documented by the published case of a patient who developed M. avium lymphadenitis in the setting of no prior CLL treatment [4]. However, the precise impact of these immunological defects on infection risk in CLL, and their reversibility after CLL treatment has been difficult to quantify [18].

The key risk factor for infectious complications in CLL seems to be the number and type of immunosuppressive agents previously used for CLL treatment [18,19,21]. For example, infectious complications may occur in the setting of neutropenia or T-cell dysfunction which may follow treatment with bendamustine, alemtuzumab, obinutuzumab, rituximab, or fludarabine [18,20]. The immunosuppressive effects of CLL therapies (e.g., alemtuzumab, fludarabine, rituximab) may be profound and prolonged, i.e., they may persist for years after therapy is discontinued. It is thus likely that the cumulative effect of different CLL treatment agents given over many years predisposed our patient to immune dysfunction and opportunistic NTM infection.

The initial presentation of infections in CLL typically includes pneumonia or bacteremia caused by encapsulated bacteria, with herpes viruses, urinary tract, central nervous system, and soft tissue infections occurring less frequently. Viral infections appear more often in patients treated with purine analogues or autologous stem cell transplantation. Infection prevention strategies in CLL should be selected on a case by case basis and may include vaccination (seasonal influenza vaccine, pneumococcal vaccine), intravenous immune globulin replacement in the setting of recurrent bacterial infections, consideration of hepatitis B antiviral treatment to prevent reactivation, and pneumocystis prophylaxis [19].

### 3.3. Obinutuzumab

Obinutuzumab is a humanized monoclonal antibody that acts as an immunomodulator by targeting CD20 on the surface of B-lymphocytes. In a randomized trial of patients with previously untreated CLL, obinutuzumab plus chlorambucil was associated with improved progression-free survival compared to rituximab plus chlorambucil [23]. Importantly, patients that received obinutuzumab more frequently had neutropenia, but the occurrence of serious infections was similar (11–14% in each study arm). Infections that occurred after treatment with obinutuzumab were mostly bacterial.

To our knowledge, ours is the first report of a mycobacterial infection following obinutuzumab administration in a CLL patient. The generalized lymphadenopathy significantly regressed after starting obinutuzumab treatment, and retroperitoneal/unilateral inguinal lymphadenopathy subsequently increased, suggesting the need for a diagnostic biopsy which led to the diagnosis of M. avium infection. Therefore, it appears unlikely that disseminated M. avium infection was already present before obinutuzumab was started. However, the significance of obinutuzumab treatment in contributing to M.avium infection in our patient is unclear, given the long history of CLL and numerous immunosuppressive treatments the patient had previously received.

### 3.4. Ibrutinib

Another target-specific agent recently introduced for treating CLL is ibrutinib [24]. It irreversibly inhibits Bruton’s tyrosine kinase, which is a key enzyme involved in the maturation and function of B-lymphocytes. Ibrutinib was given to our patient after re-appearance of generalized lymphadenopathy despite obinutuzumab treatment and following successful antimicrobial therapy of the M. avium psoas abscess. Ibrutinib may also be beneficial in patients over 65 years of age with previously untreated CLL and was recorded to be more effective regarding progression-free survival compared to chemoimmunotherapy with bendamustine plus rituximab [25].

Ibrutinib is typically well tolerated for extended periods in many patients with CLL and has not been associated with a higher infection risk than with comparator CLL treatments, neither in combination with alkylating drugs or anti-CD20 monoclonal antibody therapy nor as monotherapy in treatment-naïve CLL patients [18,22]. Occasional case reports of CLL patients with infections after ibrutinib therapy have included lower respiratory tract infections and bacterial pneumonia. Opportunistic infections have only rarely been reported after ibrutinib, and have included cases of Pneumocystis jirovecii pneumonia, disseminated fusariosis, cryptococcal meningitis, and invasive aspergillosis [18].

## 4. Conclusions

Our case and our literature review illustrate that physicians should be aware of opportunistic infections in patients with CLL.Infectious complications may be difficult to recognize clinically. Appropriate biopsy material from affected lymph nodes should be obtained. The key to diagnosis of opportunistic infection is microbiological confirmation and histological exclusion of secondary malignancies, Richter’s transformation, and progressive CLL.Diagnosis of NTM infection relies on mycobacterial PCR and culture done on biopsy material (or broncho-alveolar lavage fluid when there is respiratory involvement).Infection prophylaxis including vaccinations or intravenous immune globulin replacement should be considered in all CLL patients when appropriate.

## Figures and Tables

**Figure 1 jcm-08-00216-f001:**
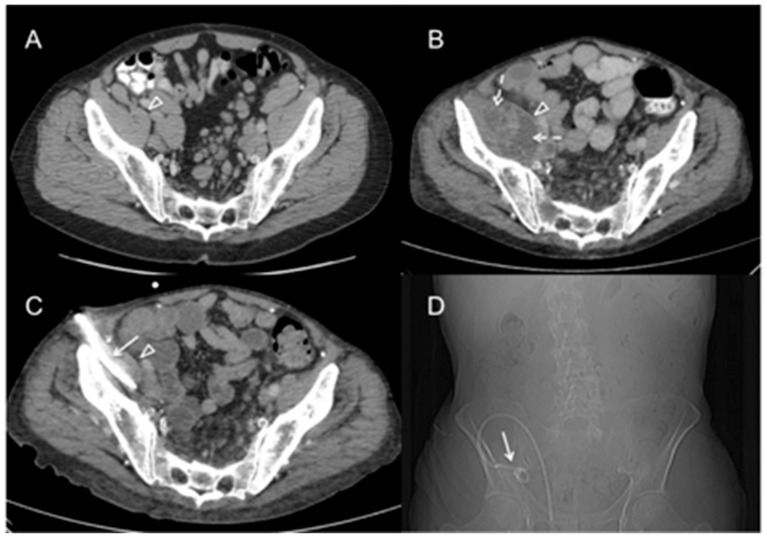
*Mycobacterium avium* psoas abscess and inguinal lymphadenitis in a patient with chronic lymphocytic leukemia (CLL). (**A**) Computed tomography with intravenous and oral contrast, done at the time of initiation of intravenous immune globulin (IVIG) replacement, demonstrates normal and symmetric psoas muscle appearance (arrowhead), without abscess formation. (**B**) Large, septated abscess (dashed arrows) located within the right psoas muscle (arrowhead), of which biopsy material grew *M. avium* in culture. (**C**) One month later, notable reduction in psoas abscess volume (arrowhead) after introduction of computed tomography (CT)-guided percutaneous pigtail catheter drain (arrow). (**D**) CT scout image corresponding to axial image C, demonstrating the course of the percutaneous pigtail catheter drain (arrow).

**Table 1 jcm-08-00216-t001:** Reported cases of *Mycobacterium avium* complex (MAC) infection in patients with CLL.

Age at CLL (year)	Age at MAC (year)	Sex	Clinical Manifestations of MAC Infection	Site of MAC Detection	CLL Treatment before MAC Diagnosis	Outcome	Ref.
60	74	Male	retroperitoneal and inguinal lymphadenitis, psoas abscess	Inguinal lymph node biopsy	chlorambucil, prednisone, fludarabine, rituximab, cyclophosphamide, bendamustin, ofatumumab, lenalidomide; intravenous immune globulin; obinutuzumab	MAC successfully treated, patient dies several months later	present case
38	46	Male	lymphadenitis	lymph node biopsy	alemtuzumab	alive	[1]
41	55	Male	lymphadenitis, small bowel infection	n/a	cyclophosphamide/oncovine/prednisone chemotherapy	alive	[2]
49	59	Male	disseminated cutaneous disease	skin biopsy	fludarabine, prednisone, cyclophosphamide, rituximab, alemtuzumab	died	[3]
n/a	n/a	Male	lymphadenitis	lymph node biopsy	none	alive	[4]

**Table 2 jcm-08-00216-t002:** Reported cases of non-tuberculous mycobacterial NTM infections other than *M. avium* in patients with CLL.

Age at CLL (year)	Age at NTM (year)	Sex	NTM Species	Clinical Manifestations of NTM Infection	Site of NTM Detection	CLL Treatment before NTM Diagnosis	Outcome	Ref.
48	58	F	*M. chelonae*	disseminated cutaneous disease	skin biopsy	methylprednisolone and other 5 non specified agents	alive	[10]
74	85	M	*M. chelonae*	skin lesions on arms and legs	skin biopsy	bendamustin, rituximab, fludarabine	alive	[11]
67	74	M	*M. chelonae*	skin lesions	skin biopsy	chlorambucil, fludarabine, cyclophosphamide, alemtuzumab	alive	[12]
62	n/a	M	*M. marinum*	erythematous cutaneous nodules, pulmonary nodule	skin biopsy	prednisolone, ciclosporin	died	[13]
57	64	M	*M. marinum*	skin lesions	skin biopsy	fludarabine, cyclophosphamide, rituximab	alive	[14]
61	62	M	*M. fortuitum*	pulmonary nodules	sputum	chlorambucile, fludarabine	died	[16]
62	79 and 83	F	*M. genavense*	lymphadenitis	blood culture and bone marrow biopsy	chlorambucil, prednisone	died	[17]
59	66	F	*M. szulgai*	multifocal osteomyelitis, cutaneous lesions	bone biopsy	fludarabine, chlorambucil	died	[15]

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
