# Peer review of "Psoas Abscess Due to Mycobacterium avium in a Patient with Chronic Lymphocytic Leukemia—Case Report and Review"

_jcm, 2019, doi:10.3390/jcm8020216_

Round 1
Reviewer 1 Report
In their manuscript ”Psoas Abscess due to Mycobacterium Avium in a Patient with Chronic Lymphocytic Leukemia – case report and review”, Diaco et al. revisit the risk of infectious complications in the era of anti-CD20 and Bruton tyrosine kinase inhibitor-based therapies of chronic lymphocytic leukaemia (CLL). They base their report on a case of an atypical mycobacterial abscess of a heavily pre-treated CLL patient, including anti-CD20 therapy, which – they argue – is (one of the) first reported case of a non-tuberculous mycobacterial infection in a patient exposed to obinutuzumab therapies in CLL.
Comments:
The authors should discuss whether the progressive retroperitoneal lymphadenopathy in Dec 2013 was CLL progression or manifestation of MTN infection, preceding the psoas abscess. This is important as to know whether overt infection was present prior to or following obinutuzumab therapy. The reviewer appreciates that this might not be answered with certainty. Related to this, it should be discussed whether previous chemotherapy – though discontinued years before manifestation of the psoas abscess – might have contributed to the acquisition/manifestion of MTN infection. This is of particular interest given the referenced report in citation no. 4 without preceding therapy. Are the authors able to present direct or indirect evidence of T-cell dysfunction in the clinical history of the patient (lymphopenia, changes to CD4/CD8 ratios, etc)? Also this is relevant given the authors only mention neutropenia as a known association with anti-CD20 therapy.
The authors want to make the point that mycobacterial infections can occur in CLL without ongoing chemotherapy as a risk factor. However, they should make more clear that CLL in itself is a risk factor, interfering with several aspects of adaptive immunity. Otherwise, the reader might be mistaken that the authors want to imply a causal relationship between novel therapies of CLL and mycobacterial infections. This is, again, of particular interest given the referenced report in citation no. 4 without preceding therapy.
Even though the authors do mention the phenocopy of mycobacterial infection and CLL progression in their conclusion, they should underline the importance of considering mycobacterial infection a true differential diagnosis of CLL progression. Furthermore, they should discuss in which cases diagnostics to exclude mycobacterial infection in CLL is indicated.
Author Response
Reviewer new to the revised manuscript version:
The authors should discuss whether the progressive retroperitoneal lymphadenopathy in Dec 2013 was CLL progression or manifestation of MTN infection, preceding the psoas abscess. This is important as to know whether overt infection was present prior to or following obinutuzumab therapy. The reviewer appreciates that this might not be answered with certainty.
Response: Agree. We therefore added 2 clarifyling sentences to section 3.3.
Related to this, it should be discussed whether previous chemotherapy – though discontinued years before manifestation of the psoas abscess – might have contributed to the acquisition/manifestion of MTN infection. This is of particular interest given the referenced report in citation no. 4 without preceding therapy. Are the authors able to present direct or indirect evidence of T-cell dysfunction in the clinical history of the patient (lymphopenia, changes to CD4/CD8 ratios, etc)? Also this is relevant given the authors only mention neutropenia as a known association with anti-CD20 therapy.
Response: We agree and have added such discussion in the revised manuscript (lines 183 and following). CD4/CD8 ratios were unfortunately not assessed in our patient.
The authors want to make the point that mycobacterial infections can occur in CLL without ongoing chemotherapy as a risk factor. However, they should make more clear that CLL in itself is a risk factor, interfering with several aspects of adaptive immunity. Otherwise, the reader might be mistaken that the authors want to imply a causal relationship between novel therapies of CLL and mycobacterial infections. This is, again, of particular interest given the referenced report in citation no. 4 without preceding therapy.
Response: We agree and have modified the respective paragraph 3.2. Infectious Complications in CLL.
Even though the authors do mention the phenocopy of mycobacterial infection and CLL progression in their conclusion, they should underline the importance of considering mycobacterial infection a true differential diagnosis of CLL progression. Furthermore, they should discuss in which cases diagnostics to exclude mycobacterial infection in CLL is indicated.
Response: We agree and have added some additional sentences to the beginning of the discussion.
Reviewer 2 Report
Improved from prior submission
Minor comments-
The section clarifying the administration of ibrutinib with respect to receipt of the anti-mycobacterial drugs is helpful
Obinutuzumab is misspelled on page 3 (line 95)
Author Response
Original Reviewer :
Improved from prior submission
Minor comments-
The section clarifying the administration of ibrutinib with respect to receipt of the anti-mycobacterial drugs is helpful
Obinutuzumab is misspelled on page 3 (line 95)
Response: This has been corrected
This manuscript is a resubmission of an earlier submission. The following is a list of the peer review reports and author responses from that submission.
Round 1
Reviewer 1 Report
This is a well written report describing a case of mycobacterium avium infection involving a retroperitoneal lymph node and right psoas muscle in an extensively pre-treated CLL patient.
This emphasizes the importance of biopsy/culture in patients with CLL and growing node(s) where disease progression is in the differential diagnosis to exclude the possibility of infection.
Indeed M.avium is an unusual infection among CLL patients though CLL patients are known to be at an increased infection risk especially those with multiple lines of prior therapy.
I think that the clinical case description requires more details to make the case clearer to the reader.
A few points:
1) I could be wrong but my general idea was that specific dates should not be included in case reports to protect patient privacy; could instead change dates ("May 2014") to general intervals ("5 months later")
2) did the patient receive all planned doses of obinutuzumab or was therapy stopped due to infection
3) was the patient still on the rifampicin/ethambutol/clathromycin regimen when ibrutinib was initiated, and if so, I would suggest authors discuss the drug-drug interactions that may have played a role in the patient's subsequent course (clarithromycin can increase ibrutinib levels; rifampin the opposite - the combination likely leading to unclear/unpredictable pharmacokinetics)
4) Lastly, I think the emphasis on the patient being the first to have M.avium following obinutuzumab is perhaps a bit exaggerated; as this patient's immunosuppression is likely multifactorial but is primarily driven by his extensive, extensive prior therapies +/- hypogammaglobulinemia which could be due to prior therapy + disease. Next, the concept that it is the first reported/published case following obi is more due to obi being a new drug as opposed to this being a meaningful association.
Author Response
A few points:
1) I could be wrong but my general idea was that specific dates should not be included in case reports to protect patient privacy; could instead change dates ("May 2014") to general intervals ("5 months later")
Answer: Agree. All dates have been removed, i.e. converted to general intervals, in the case report section and in the legend to Figure 1.
2) did the patient receive all planned doses of obinutuzumab or was therapy stopped due to infection
Answer: This information has been added to the revised manuscript; the second cycle of obinotuzumab was delayed by three months due to the infectious complication. The third cycle was started on time but terminated early due to the progression of the CLL.
3) was the patient still on the rifampicin/ethambutol/clathromycin regimen when ibrutinib was initiated, and if so, I would suggest authors discuss the drug-drug interactions that may have played a role in the patient's subsequent course (clarithromycin can increase ibrutinib levels; rifampin the opposite - the combination likely leading to unclear/unpredictable pharmacokinetics)
Answer: We agree and have re-written this passage. The patient was still on the rifampicin/ethambutol/clathromycin regimen when ibrutinib was started. The initial dose of ibrutinib was reduced to 280 mg due to low blood counts at start and the risk of potential interactions with increased blood levels of ibrutinib. The second and third cycle were started with 420 mg for 2 weeks and reduced to 280 mg for the rest of the cycle due to low platelet counts. Later cycles were given at the standard dose for CLL, i.e. 420 mg without enhanced hematotoxicity despite the combination with the antibiotics. Blood level measurements for ibrutinib were not done.
4) Lastly, I think the emphasis on the patient being the first to have M.avium following obinutuzumab is perhaps a bit exaggerated; as this patient's immunosuppression is likely multifactorial but is primarily driven by his extensive, extensive prior therapies +/- hypogammaglobulinemia which could be due to prior therapy + disease. Next, the concept that it is the first reported/published case following obi is more due to obi being a new drug as opposed to this being a meaningful association.
Answer: Agree. The abstract, introduction, and results have been rephrased using more cautious language.